# Developmental Trajectory of Anticipation: Insights from Sequential Comparative Judgments

**DOI:** 10.3390/bs13080646

**Published:** 2023-08-03

**Authors:** Leslie Tricoche, Martine Meunier, Sirine Hassen, Jérôme Prado, Denis Pélisson

**Affiliations:** 1IMPACT Team, Lyon Neuroscience Research Center, University Lyon, UCBL, UJM, INSERM, CNRS, U1028, UMR5292, F-69000 Lyon, France; martine.meunier@inserm.fr (M.M.); sirine.hassen.etu@univ-lille.fr (S.H.); denis.pelisson@inserm.fr (D.P.); 2EDUWELL Team, Lyon Neuroscience Research Center, University Lyon, UCBL, UJM, INSERM, CNRS, U1028, UMR5292, F-69000 Lyon, France; jerome.prado@univ-lyon1.fr

**Keywords:** RT distribution, strategies, development, mixture model, diffusion model, sequential stimuli

## Abstract

Reaction time (RT) is a critical measure of performance, and studying its distribution at the group or individual level provides useful information on the cognitive processes or strategies used to perform a task. In a previous study measuring RT in children and adults asked to compare two successive stimuli (quantities or words), we discovered that the group RT distribution was bimodal, with some subjects responding with a mean RT of around 1100 ms and others with a mean RT of around 500 ms. This bimodal distribution suggested two distinct response strategies, one reactive, the other anticipatory. In the present study, we tested whether subjects’ segregation into fast and slow responders (1) extended to other sequential comparative judgments (2) evolved from age 8 to adulthood, (3) could be linked to anticipation as assessed using computer modeling (4) stemmed from individual-specific strategies amenable to instruction. To test the first three predictions, we conducted a distributional and theoretical analysis of the RT of 158 subjects tested earlier using four different sequential comparative judgment tasks (numerosity, phonological, multiplication, subtraction). Group RT distributions were bimodal in all tasks, with the two strategies differing in speed and sometimes accuracy too. The fast strategy, which was rare or absent in 8- to 9-year-olds, steadily increased through childhood. Its frequency in adolescence remained, however, lower than in adulthood. A mixture model confirmed this developmental evolution, while a diffusion model corroborated the idea that the difference between the two strategies concerns anticipatory processes preceding decision processes. To test the fourth prediction, we conducted an online experiment where 236 participants made numerosity comparisons before and after an instruction favoring either reactive or anticipatory responses. The results provide out-of-the-lab evidence of the bimodal RT distribution associated with sequential comparisons and demonstrated that the proportions of fast vs. slow responders can be modulated simply by asking subjects to anticipate or not the future result of the comparison. Although anticipation of the future is as important for cognition as memory of the past, its evolution after the first year of life is much more poorly known. The present study is a step toward meeting this challenge. It also illustrates how analyzing individual RT distributions in addition to group RT distributions and using computational models can improve the assessment of decision making cognitive processes.

## 1. Introduction

Reaction time (RT), the time individuals take to initiate a response, and the accuracy of that response, are the two main measures of performance used in behavioral studies. Most studies report, however, the mean (or median) RT, sometimes with its spread (variability), neglecting group and individual RT distributions [1,2]. Yet, RT distributions have helped us understand the effects of conditions such as attention-deficit/hyperactivity [3,4,5,6], as well as healthy variations in cognition such as the effects of age across the lifespan [7,8,9]. Computational models (e.g., ex-Gaussian, drift diffusion, mixture, [10,11,12]) fitted to raw distributional data allow, in addition, for theoretical inferences about the cognitive processes responsible for RT changes [13,14]. Together, distributional and theoretical analyses of RT have also been instrumental in tracking response strategies and their variations across and within individuals [9,15].

RT tasks generally propose carrying out a comparison of two stimuli (S1 and S2) before making a two-alternative choice with the instruction to be both fast and accurate. They therefore leave two possible types of strategies: reactive and anticipatory strategies [16]. Reactive strategies rely exclusively on events having already occurred to produce cautious accurate responses. Anticipatory strategies rely on events that have not yet occurred to produce fast risky responses. When S1 and S2 appear simultaneously (side-by-side), anticipation provides only fast guesses leading to chance-level performance [15,16]. In such cases, a “wait and see” reactive approach may therefore be the most adaptive strategy [9]. By contrast, when S1 and S2 are presented sequentially (one after the other), anticipation may be more adaptive. Subjects can, in this case, use S1 as a preparatory cue to preprogram one response during the delay, thus reducing S2 processing to a mere win–stay lose–shift decision [16]. Husain and his colleagues termed this capacity to make rapid decisions and negotiate risk early “functionally-useful anticipation” [9].

However useful, anticipation of the future is often overlooked in cognitive psychology. Yet, it occupies our spontaneous thoughts as much as memory of the past [17]. It possesses neural correlates, such as the anticipatory lateralization of visual alpha rhythms, capable of predicting trial-by-trial fluctuations in behavioral responses as indexed by RT [18]. Even more critical, improvement in anticipatory skills is a central pillar of development thought to predict later cognitive abilities, including language and executive functions [19]. Anticipation development has nevertheless been mainly studied in infancy [20]. Unlike memory development, whose brain and behavior correlates have been thoroughly and meticulously mapped throughout the lifespan, the developmental trajectory of anticipation through childhood has yet to be elucidated. The present behavioral study is part of this effort. It uses sequential comparative judgments as a window into anticipatory capacities, and distributional and theoretical analysis of RT as a tool to track their evolution with age.

In a recent study, we tested sequential numerosity and phonological judgments in adults and children aged 10 to 13 [21]. Two arrays of dots or two written words were successively presented. Two response keys were available for the subject to tell which of the two arrays had more dots, or whether the two words rhymed or not. We found that some subjects showed a mean RT of around 1100 ms, while others managed to respond no less than 600 ms earlier, showing a mean RT of around 500 ms. This bimodal shape of the group RT distribution, indicative of the presence of two distinct strategies, one reactive, the other anticipatory, was present in both adults and children. The difference was that the proportion of fast responders was much smaller in children. These results led us to speculate that there is a developmental effect in strategy preference, with adults having reached a better capability than children to use the anticipatory strategy.

In the present study, we tested (1) if the segregation between fast and slow responders described above is generalizable to other sequential comparative judgments, (2) how it evolves from the age of 8 up to adulthood, (3) whether inferring a role of anticipation is supported by evidence from computer modeling and (4) whether it stems from individual-specific strategies amenable to instruction. These objectives were addressed in two experiments. Experiment 1 traced the trajectory of anticipation through childhood, adolescence and early adulthood. To this aim, it re-analyzed behavioral data collected earlier in 158 participants, including 132 8- to 15-year-olds and 26 young adults, performing four types of comparisons between two sequential stimuli (S1 and S2) during fMRI brain imaging (see [22,23,24,25,26] for detail). In numerosity judgments, participants decided whether the S1 or S2 dot array had the largest number of dots. In phonological judgments, they decided whether S1 and S2 words rhymed. In multiplication judgments, they decided whether S1 single-digit multiplication led to the result given in S2. In subtraction judgments, they decided whether S1 single-digit subtraction led to the result given in S2. Based on the study of Tricoche et al., 2021, we expected a bimodal group RT distribution revealing a segregation between slow and fast responders, confirmed by a mixture model, with the proportion of fast responders increasing with age across all four tasks. We also expected the diffusion model to provide theoretical evidence that the difference between the two strategies lies in pre-decision (anticipatory) processes rather than decision (reactive) processes. Experiment 2 assessed whether the segregation between fast and slow responders would be sensitive to a change in task instruction. It took advantage of the large sample sizes provided by online testing, namely 236 adults tested on two successive blocks of numerosity judgments. Whereas no instruction was given regarding a strategy to be used in the first block, the second block was preceded by an instruction nudging subjects into either anticipatory or reactive responses. We hypothesized that compliance with the instruction would trigger abrupt and massive RT changes indicating a strategic adjustment and ruling out any gradual mechanism such as learning.

## 2. Materials and Methods—Experiment 1

### 2.1. Dataset and Participants

The behavioral datasets were previously acquired in 132 child participants ranging in age from 8 to 15 (mean = 11.26 +/− 1.46) and in 26 adults ranging in age from 19 to 30 (mean = 25.19 +/− 3.07). Datasets are available on OpenNeuro [27,28], and experimental procedures were approved by the Northwestern University Institutional Review Board. All participants were (1) recruited and tested in the Chicago area (USA), (2) native English speakers and (3) healthy participants without any psychiatric or neurological disorders. Individual performance together with each subject’s verbal Intelligence Quotient (IQ), cognitive abilities (reading ability, symbolic math performance, math fluency, basic calculation, visuospatial working memory) and Socioeconomic Status (SES; based on the parental education level for children), are detailed in the original studies (see, e.g., [22,23,24,25,26]). Participants had standard performance and verbal IQ scores, and no intellectual, mathematical or reading impairment.

To assess developmental differences, we divided the child population into three age subgroups of comparable size: 8- to 9-year-old children (N = 31), 10- to 11-year-old children (N = 57) and 12- to 15-year-old children (N = 44).

### 2.2. Tasks

Participants completed four tasks in a counterbalanced order: numerosity judgment, phonological judgment (rhyme), multiplication and subtraction. All participants completed these tasks during an MRI scanning session, but in the present study we only analyzed behavioral data.

Numerosity judgment (Figure 1A). Participants had to decide which of two dot arrays presented successively contained more dots. The ratio of the number of dots between the two arrays was 0.33 (i.e., 12 vs. 36 dots) for “simple trials”, 0.5 (i.e., 18 vs. 36 dots) for “medium trials” and 0.66 (i.e., 24 vs. 36 dots) for “complex trials”. Participants completed 24 trials for each difficulty level (totalizing 72 trials presented randomly). 

Phonological judgment (Figure 1B). Participants had to decide whether two successively presented words rhymed or not. For simple trials, phonology and spelling were congruent (identical phonology and identical orthography, e.g., dime–lime, or different phonology and different orthography, e.g., press–list), whereas for complex trials, phonology and spelling were incongruent (different phonology but identical orthography, e.g., pint–mint, or identical phonology but different orthography, e.g., jazz–has). Participants completed 24 trials for each difficulty level (totalizing 48 trials presented randomly). 

Multiplication (Figure 1C). A single-digit multiplication problem and a number were presented successively. Participants had to decide whether or not the number was the correct result of the multiplication. For simple trials, the two operands of the multiplication were smaller than or equal to 5 (e.g., 3 × 2), whereas for complex trials, the two operands were larger than 5 (e.g., 6 × 8). Participants completed 72 trials (36 simple, 36 complex) presented randomly. 

Subtraction (Figure 1D). A single-digit subtraction problem and a numberwere presented successively. Participants had to decide whether or not the number was the correct result of the subtraction. For simple trials, the difference between the two terms of the subtraction was smaller than 3 (e.g., 4 − 2), whereas for complex trials, the difference between the two terms was larger than 3 (e.g., 8 − 2). Participants completed 72 trials (36 simple, 36 complex) presented randomly.

### 2.3. Trial Sequence

As illustrated in Figure 1, for each trial, the two stimuli (two dot arrays for numerosity judgment, two words for rhyme judgment, a single-digit multiplication problem and a number for multiplication, a single-digit subtraction problem and a number for subtraction) appeared one after the other for 800 ms each, with a 200 ms delay in between. A red square then appeared for a duration varying randomly from 2800 ms to 3600 ms. Participants had to decide which array contained the largest number of dots (numerosity judgment) or whether the two words rhymed or not (rhyme judgment) or if the multiplication problem’s (or subtraction problem’s) proposed result (number) was correct or not. They were asked to respond as fast and accurately as possible by pressing a key as soon as the second stimulus appeared and before the red square turned off. For a full description of the tasks and protocols, see notably Berteletti and Booth (2015) [23] and Prado and collaborators (2011, 2014) [25,29].

### 2.4. Analyses

All analyses were conducted using R (RStudio, v.1.0.136) on the same two measures for all participants and all four tasks: accuracy (=percentage of correct responses) and RT (=time between S2 onset and the key press). For each task, we conducted descriptive and statistical analyses of correct RT distributions at the group and individual level. Because, as predicted, individual RT distributions were in most subjects unimodal (indicative of a single strategy) and group distributions were generally bimodal (indicative of two strategies), we classified each participant as either a slow or a fast responder, and then calculated the proportions of subjects for each response profile (fast, slow) separately for each age group and each task. The segregation between slow and fast responders was carried out in relation to the latency of the group distribution trough (lowest density point between the two peaks of the bimodal distribution as defined by the R software density function, i.e., approximatively 1100 ms for adults in all 4 tasks). Subjects with RT peaks superior to the trough latency were classified as “slow responders”; subjects with RT peaks inferior to the trough latency were classified as “fast responders”. For the 8- to 9-year-olds, as the group distributions for the numerosity and subtraction tasks were not bimodal but, in contrary, unimodal with a peak around 1100 ms, all participants were classified as “slow responders” in these two tasks. We also controlled for an effect of difficulty level (easy, hard) and analyzed its interaction with the response profile, because task difficulty is known to modulate performance and speed [30,31]. For this analysis, each participant was classified into either the fast or slow responder group for each difficulty level (note that the intermediate level of the numerosity task, which had no counterpart in the other tasks, was removed from the analysis). In conducting a Cochran–Mantel–Haenszel test (CMH χ^2^, with the continuity correction) with response profile, difficulty level and age as independent variables, we did not find changes in the relative proportion of participants showing a fast or slow response profile with the difficulty level when controlling for age (X^2^ = 0.18, *p* = 0.67). Consequently, the difficulty level factor was removed from all subsequent analyses.

A 4 × 4 × 2 ANOVA with age (8–9 years, 10–11 years, 12–15 years and adults) as between-subject factor, task (numerosity, rhyme, multiplication and subtraction) as within-subject factor and response profile (fast, slow) as between- or within-subject factor (depending on whether participants changed their response profile across tasks) was conducted on the percentage of correct responses to identify the potential speed–accuracy trade-off between slow and fast responders. Separated age x response profile ANOVAs for each task were also conducted. As the percentages of correct responses were not normally distributed, we log-transformed the data before conducting the ANOVA in order to fit them with our negatively skewed distribution: log-transformed %correct = log10 (max (%correct + 1) − %correct). Significant interactions were followed up with Bonferroni- or FDR-corrected pairwise comparisons. Effect sizes were reported as partial eta-squared values (η_p_^2^) for each ANOVA. η_p_^2^ = 0.01 indicates a small effect, η_p_^2^ = 0.06 indicates a medium effect, and η_p_^2^ = 0.14 indicates a large effect [32]. 

We conducted CMH χ^2^ with response profile, task and age as independent variables. It allowed us to determine whether the proportion of participants changed as a function of response profile, age and task. Chi-square tests were also applied when the CMH χ^2^ test showed significant results. Additional post hoc comparisons were conducted when needed. Cramer’s V was calculated to estimate the effect sizes for each chi-square test. A commonly used interpretation is to qualify effect sizes as small for V > 0.06, medium for V > 0.17, and large for V > 0.29 (with our degree of freedom being equal to 3) [33].

To corroborate the empirical analysis of the segregation between fast and slow responders described above, we conducted a theoretical analysis using the mixture model [11,34,35]. This probabilistic model identifies the subpopulations composing a group distribution by defining a separate Gaussian distribution for each of them. For each task and each age group, the two mixture model Gaussians were fitted to RT distribution with the mean (mu parameter) set to the mean adult RT for each strategy, and the variance (sigma parameter) left free. For each resulting Gaussian sub-distribution, a lambda parameter was estimated. That parameter corresponded to the probability that a RT taken randomly from the actual group distribution was part of this Gaussian distribution. This lambda parameter was submitted to a CMH χ^2^ test and the associated chi-square and post hoc tests, allowing us to assess whether independent factors (response profile, age and task) influence the probability of belonging to one or the other Gaussian sub-distribution.

Finally, we used a diffusion model to provide theoretical inferences about the cognitive processes differing across the slow and fast strategies. The model includes RT for both correct and incorrect responses. It estimates two decision parameters and one non-decision parameter per participant. The first decision parameter is the drift rate (v), which estimates how quickly and efficiently an individual can accumulate information to inform his/her response decision. The second decision parameter is the boundary separation or threshold (a) corresponding to the accumulated information level a person needs to commit to a response (certainty level). The non-decision parameter (t0) aggregates all the processes preceding the decision, including maintenance of the stimuli in working memory and preparation of the motor response. We fitted the model following exactly the same method as the one we used in Tricoche et al., 2021 [21] (please refer to this paper for more information about the procedure). Each of the 3 estimated model parameters (a, v and t0) was submitted to a three-way 4 × 4 × 2 ANOVA with age as the between-subject factor, task as the within-subject factor and response profile as the between- or within-subject factor depending on whether participants changed their response profile between tasks. Effect sizes were reported as partial eta-squared values (η_p_^2^) for each ANOVA.

The significance level was set at *p* < 0.05 for all analyses.

## 3. Results—Experiment 1

In this section, we provide for each task a descriptive analysis of the RT distributions at the group level to assess their shape (uni- or bimodal) and to characterize their developmental changes. We then plot for each task all individual RT distributions in order to classify each participant as a “slow” or a “fast” responder. We analyze whether this classification evolves across ages, particularly using CMH χ^2^ tests. We also check if the two responder’s profiles are associated with different levels of response accuracy (percentage of correct responses). Finally, the mixture model is fitted to the data to consolidate the results and notably the developmental course of the response strategy.

### 3.1. Group-Level Distributions

A table summarizing the mean and variance of RTs at group level for each age, task and difficulty level is given in Appendix A. RT distributions at the group level are plotted in Figure 2 for each task (across difficulty levels) and for each age group. This figure confirms the presence of a bimodal distribution for each task, but with some differences between the age groups. For all ages and all tasks, a first RT peak appeared around 600 ms (except for the 8- to 9-year-old children, who did not show this first peak for the numerosity and subtraction tasks) and a second peak around 1100 ms, with a trough latency around 900 ms. This trough latency was measured for each age group in each task as the lowest density point between the two unimodal sub-distributions. The relative density in each peak, however, changed with age: the second peak was highest during childhood but decreased progressively in favor of the first peak, which became largely predominant in adults.

### 3.2. Individual Distributions

Participants were characterized as fast or slow responders depending on whether their individual RT distribution peaked at a latency shorter (fast responder) or longer (slow responder) than the population trough latency. As shown in Figure 3, almost all 8–9-year-old children were classified as slow responders (orange distributions). The number of fast responders (blue distributions) increased from 10–11 to 12–15 years of age without reaching adult level. Indeed, nearly all adults were fast responders.

We then calculated the proportion of fast versus slow responders for each age and task (Figure 4), and we conducted CMH χ^2^ tests to evaluate the effects of age, task and response profile on the proportion of participants. We first investigated if there was an association between response profile and task that was dependent upon age. We found that the relative proportion of participants showing a fast or slow response profile significantly changed with the task when controlling for the age (CMH: M^2^ = 21.29, *p* < 0.001). Specifically, follow-up chi-square tests revealed a significant response profile × task interaction in 8- to 9-year-old children (X^2^ = 18.37, *p* < 0.001, V = 0.12), in 10- to 11-year-old children (X^2^ = 8.71, *p* = 0.03, V = 0.08) and in adults (X^2^ = 8.2, *p* = 0.04, V = 0.08), but not in 12- to 15-year-old children (X^2^ = 3.54, *p* = 0.31, V = 0.05). According to Cramer’s V, the effect sizes were small for this interaction [33]. Post hoc tests showed a higher proportion of fast responders in multiplication than in the other tasks among 8- to 9- and 10- to 11-year-old children (8–9 years: *p* < 0.001; 10–11 years: *p* = 0.05). In adults, post hoc tests pointed to a non-significant trend toward a higher proportion of fast responders in the multiplication and subtraction tasks than in the numerosity and rhyme tasks.

We then investigated if the association between response profile and age was dependent upon the task. This time, the relative proportion of fast versus slow responders significantly changed with the age when controlling for the task (CMH: M^2^ = 257.18, *p* < 0.001). Specifically, follow-up chi-square tests revealed a significant response profile × age interaction in all four tasks (numerosity: X^2^ = 61.9, *p* < 0.001, V = 0.22; rhyme: X^2^ = 61.75, *p* < 0.001, V = 0.23; multiplication: X^2^ = 48.67, *p* < 0.001, V = 0.20; subtraction: X^2^ = 93.37, *p* < 0.001, V = 0.28). For all tasks, Cramer’s V estimated this interaction as a medium effect higher than 0.20, indicating that age relatively strongly influences the proportion of fast and slow responders. All post hoc tests indicated that the two youngest groups of children (8- to 9-year-olds and 10- to 11-year-olds) had fewer fast responders than adults (all *p*’s < 0.05 except numerosity for 10–11-year-olds, where *p* = 0.06), whereas the 12- to 15-year-old children did not differ from adults (all *p*’s = 1). Indeed, there was a similar proportion of fast and slow responders among 12- to 15-year-old children, the intermediate profile between the profile of the two youngest child groups and the profile of adults.

Finally, we calculated the proportion of participants who did not follow the same strategy in all four tasks (i.e., participants who showed a strategy shift in at least one of the four tasks). In the three child groups, one-third of the participants did not have a consistent strategy in all tasks (8–9 years: 36.7%; 10–11 years: 33.9%; 12–15 years: 31.8%). In contrast, adults were more consistent than children, as only 9.6% of them changed between fast and slow responders across tasks.

### 3.3. No Main Effect of Response Profile on Accuracy

A table summarizing the mean and variance of accuracy at group level for each age, task and difficulty level is given in Appendix A. To evaluate whether response profile impacted accuracy, we applied to the log-transformed percentage of correct responses a three-way ANOVA with the factors age as the between-subject factor, task as the within-subject factor and response profile as the between- or within-subject factor depending on whether participants changed their response profile between tasks. We found a main effect of age (F (3161) = 8.79, *p* < 0.001, η_p_^2^ = 0.14), revealing an expected improvement in performance with age. A main effect of task was also present (F (3519) = 134.74, *p* < 0.001, η_p_^2^ = 0.41), such that accuracy was higher for numerosity (mean = 96.43%, E-C = 5.12) and rhyme (mean = 95.47%, E-C = 6.55) than for multiplication (mean = 84.48%, E-C = 13.28) and subtraction (mean = 86.35%, E-C = 14.59). An age x task interaction (F (9519) = 43.11, *p* < 0.001, η_p_^2^ = 0.43) further indicated that the improvement with age was particularly potent for multiplication and subtraction (two skills that are arguably acquired later in children than numerosity comparison and rhyme comparison). 

More central to our current interests, the main effect of response profile was also significant when considered a within-subject factor (F (1519) = 10.13, *p* = 0.001, η_p_^2^ = 0.02) and almost significant when considered a between-subject factor (F (1161) = 3.57, *p* = 0.06, η_p_^2^ = 0.02), being associated with a response profile x age interaction (F (3519) = 4.39, *p* = 0.005, η_p_^2^ = 0.02). Slow responders were more accurate than the fast responders, but this effect seemed to be reduced with age (Figure 5). Post hoc analyses revealed that this difference mostly occurred in 10–11-year-old children, but reached significance only when FDR correction was used (Bonferroni: *p* = 0.09, FDR: *p* = 0.02).

To complement these observations, we also submitted response accuracy to a two-way age x response profile ANOVA, separately for each task. The main effect of response profile was only found for numerosity, showing again that slow responders were more accurate than fast responders (F (1176) = 4.78, *p* = 0.03, η_p_^2^ = 0.01). No age × response profile interaction was found for any task (all *p*’s > 0.05). Overall, these results indicate a speed–accuracy trade-off where being a fast responder was associated with lower accuracy. However, this effect (which was particularly found in numerosity) seemed to mainly occur in children. This suggests that adults were able to adopt an optimized strategy with a performance that was both fast and accurate.

### 3.4. Mixture Model

The mixture model was used to characterize the two components of group RT distributions using two separate Gaussian distributions. The number (two) of Gaussian distributions was constrained in our modeling, as well as their mean, which was set to the adult group’s distribution mean of each strategy, but the variance was a free parameter. As shown in Figure 6, the modeling found two Gaussian distributions with associated lambda values (see the following paragraph), except for the 8- to 9-year-old children in rhyme and subtraction, where only the long latency Gaussian was disclosed. This indicates a zero probability of being a fast responder in these two conditions. These modeling results suggest that slow and fast responders do use separate decision processes, confirming our classification results based on the raw data using the trough of population-level RT distribution (see above). 

To quantify the classification between fast or slow responders, we analyzed the lambda value associated with each fitted Gaussian. This value reflects the probability that a given RT taken randomly from the actual group distribution belongs to one of the fitted Gaussian distributions, i.e., belongs to the fast or slow responder category. The lambda values (represented in Figure 7) were analyzed using CMH χ^2^ tests with age, task and response profile as factors. As with the previous CMH χ^2^ test, the results show a significant change in lambda values according to response profile and task when controlling for age (CMH: M^2^ = 14.41, *p* = 0.002). Follow-up chi-square tests revealed a significant response profile × task interaction in all four age groups (8–9 years: X^2^ = 11.08, *p* = 0.01, V = 0.09; 10–11 years: X^2^ = 13.3, *p* = 0.004, V = 0.10; 12–15 years: X^2^ = 11.78, *p* = 0.008, V = 0.10; adults: X^2^ = 13.63, *p* = 0.003, V = 0.11). Effect sizes were categorized as a small effect for this interaction. For 8- to 9-year-old children, post hoc tests suggested a marginal, not significant, difference between the (higher) probability participants would respond quickly in numerosity compared to in the other tasks (*p* = 0.08). For both 10- to 11- and 12- to 15-year-old children, post hoc tests indicated a significantly lower probability of being a fast responder for subtraction than for the three other tasks (10–11 years: *p* = 0.005; 12–15 years: *p* = 0.02). Finally, for adults, post hoc tests showed a significantly lower probability of being a fast responder for rhyme (~50%) than for the other tasks (*p* = 0.006).

When CMH χ^2^ tests were used with response profile, age and task as factors, we found a significant change in lambda values according to response profile and age when controlling for the task (CMH: M^2^ = 503.73, *p* < 0.001). Follow-up chi-square tests revealed a significant response profile × age interaction in all four tasks (numerosity: X^2^ = 92.69, *p* < 0.001, V = 0.28; rhyme: X^2^ = 80.78, *p* < 0.001, V = 0.26; multiplication: X^2^ = 149.67, *p* < 0.001, V = 0.35; subtraction: X^2^ = 215.68, *p* < 0.001, V = 0.42), with medium or large effect sizes according to Cramer’s V. For numerosity and multiplication, post hoc tests indicated a difference between adults and both 8- to 9-year-old and 10- to 11-year-old children (numerosity: *p* < 0.001 for 8–9 years and *p* = 0.02 for 10–11 years; multiplication: *p* < 0.001 for both 8–9 and 10–11 years) but not between adults and 12- to 15-year-old children (numerosity: *p* = 1; multiplication: *p* = 0.55). For rhyme, post hoc tests showed a difference between 8- to 9-year-old children and all other age classes (*p* < 0.001), as well as a difference between adults and other age classes (*p* < 0.001). Finally, for subtraction, all age classes differed from each other (all *p*’s < 0.01).

### 3.5. Diffusion Model

We conducted an age × task × response profile ANOVA for each parameter estimated by the model (a, v and t0). A modulation of “a” could be interpreted as a change in the threshold that must be reached to make a decision, a modulation of “v” as a change in the speed of information accumulation toward that decision threshold and a modulation of “t0” as a change in pre-decision processes including preprogramming of one motor response. We found a main effect of response profile on “t0” only (as a within-subject factor: F (1515) = 259.33, *p* < 0.001, η_p_^2^ = 0.33; as a between-subject factor: F (1158) = 430.98, *p* < 0.001, η_p_^2^ = 0.73), associated with a significant triple interaction of age × task × response profile (F (9515) = 3.16, *p* = 0.001, η_p_^2^ = 0.05). Post hoc analyses indicated that t0 was shorter for fast than slow responders for all tasks and all ages (all *p*’s < 0.05), except for adults in the subtraction task, where the same tendency did not reach significance (which, however, could be explained by the weak proportion of slow adults in this task). We also found, for the “a” parameter, significant interactions between age and response profile (F (3515) = 2.82, *p* = 0.04, η_p_^2^ = 0.02) and between task and response profile (F (3515) = 3.58, *p* = 0.01, η_p_^2^ = 0.02). Follow-up analyses revealed that the difference between fast and slow responders reached significance only in adults (*p* = 0.05) and only for multiplication (*p* = 0.04) and subtraction (*p* < 0.001). The “a” parameter was higher for fast adults (1.28) than slow adults (1.02), whereas it was lower for fast responders than slow responders in multiplication (fast: 1.46, slow: 1.62) and subtraction (fast: 1.35, slow: 1.70). No main effect nor interaction was found for the “v” parameter.

## 4. Discussion—Experiment 1

The aim of the present experiment was to (1) assess whether the differences in strategies found in our previous study of numerosity and phonological comparisons [21] are generalizable to other sequential tasks (multiplication and subtraction) and whether they vary as a function of development (8–9-year-old children, 10–11-year-old children, 12–15-year-old adolescents and adults) by using a descriptive classification method and a probabilistic mixture model, and (2) investigate the underlying process differently involved between the two response profiles, fast and slow, by fitting the diffusion model to the data.

### 4.1. Two Response Profiles

Confirming our previous behavior-only study using similar numerosity and rhyme comparison tasks [21], analyses of RT distributions revealed, in all four tasks, two profiles of participants: fast responders (with a mean RT of around 500 ms) and slow responders (with a mean RT of around 1000–1100 ms). Importantly, the vast majority of participants showed a unimodal RT distribution peaking either in the fast responder range or in the slow responder range. These results suggest that participants used two different strategies.

Moreover, being a fast responder was associated with a loss in accuracy, suggesting a speed–accuracy trade-off, even if this effect seemed to decrease with age, finally leading to an optimal strategy in adults where performance was both fast and accurate. This small speed–accuracy trade-off was expected according to the findings from other studies [11,35] in which individuals who respond slowly are typically more accurate than those who respond quickly. The reason why participants chose one or the other strategy in the present study was difficult to determine from our results. However, our findings do suggest some individual specificity in strategic adjustment to two-choice tasks where S1 and S2 appear sequentially. Indeed, individuals were likely to adopt the same (fast or slow) strategy (90% of adults and 70% of children in our protocol) for any task using this specific sequential protocol.

Using the diffusion model, a model mostly applied to link speed performance and cognitive abilities at the inter-individual level [36,37,38,39], we found that the 500 ms gap between the two response profiles was predominantly driven by non-decision processes whatever the task and the age, thereby replicating the preliminary results found in our previous study [21]. Non-decision processes aggregate pre-decision working memory and motor preparation. The delay between S1 and S2 in the present comparisons provided subjects with the opportunity to anticipate their future response and preprogram it. The specific modulation found on the decision threshold in adults, and in the multiplication and subtraction tasks, suggested another more specific task-dependent process, which developed with age. Together, these modulations of decision and non-decision processes allow individuals (with their own abilities) to perform the task by optimizing their performance.

### 4.2. Developmental Differences

Our results also show that the capability of choosing the optimal strategy developed with age. In 8- to 9-year-old children, we only found slow responders, suggesting that the “functionally-useful anticipation” proposed by Husain and collaborators had not yet developed [9]. The relative proportion of fast responders then increased progressively with age with an equal number of fast and slow responders in the age range of 12–15 years. Applying the mixture model to our data confirmed the above observations by showing that the probability of belonging to the fast responder category increased with age in all tasks. Previous studies have already shown developmental improvement in strategic adjustment to informed vs. non-informed situations and win–stay vs. lose–shift decisions [40,41]. To our knowledge, however, the two response strategies shown here have not been previously identified in children. Mixture models in particular have previously been used to explain intra-individual differences where individual RT distributions were composed of a mixture of rapid stimulus detection responses and slower detection responses according to the task complexity [11,42]. Interestingly, in the present study, we found a similar distinction, but this was at the inter-individual level.

Our previous study [21] did show that adult participants could be classified as fast responders or slow responders, with a balanced 50/50 ratio between these two categories. In contrast, the proportion of fast responders in the present adults was massive, around 90%. This difference is likely due to the influence of the experimental context on behavioral performance and strategy [43,44,45]. For example, we found in our previous study that the social context influenced which strategy was used by participants [21]. Specifically, the mere presence of a partner during the task increased the ratio of fast responders. Likewise, it is important to note that the data analyzed here were collected during fMRI studies. Anxiogenic fMRI scanning tends to dissuade anxious research volunteers [46], thus potentially leading to an over-representation of competitive high performers [47]. This could explain the very high proportion of fast responders among the present participants.

Based on the literature [8], it is expected that progressively older children would develop the ability to respond as quickly and accurately as adults. Moreover, as already indicated above, a simple change in social context can help even young children to adopt the fast strategy. Moreover, confirming a previous study [48], our adult participants were quite consistent in the strategy they used across tasks. In contrast, the three child groups were more variable, as approximately one-third of them changed their strategy across tasks. These observations further suggest an immaturity of children’s ability to act as optimized fast responders. Thus, developmental and contextual factors may improve the ability of participants to adopt the most adaptive anticipatory strategy instead of keeping a cautious “wait and see” approach.

## 5. Materials and Methods—Experiment 2

### 5.1. Pilot Study

We first performed a pilot study to investigate the type of strategy used by participants and to choose the rules for Experiment 2. The eight adult participants had to first perform the numerosity and rhyme tasks, and then complete a questionnaire evaluating how they performed the tasks (e.g., “Did you adopt a specific strategy to perform the task? If yes, which one?”, “Did you think a lot of time to compare the two stimuli?”, “Did you already have an idea of your answer after the first stimulus’s appearance?”, “Did you bet on the answer before the second stimulus appearance?”, “Sometimes, did you change your mind after seeing the second stimulus?”, etc.). The results suggest that fast responders preprogrammed their future response in between S1 and S2, whereas slow responders waited for the onset of S2 to select and execute their response. Based on these preliminary results, we adapted the numerosity task for online testing. We chose this task because Experiment 1 showed that it was associated with the largest proportion of slow responders in adults, which may lend an advantage in optimizing the comparison between fast and slow responders.

### 5.2. Participants

All participants were recruited via posts on social media including a link to the task. A total of 236 participants (mean age = 31.03 +/− 11.88 years, age range = 9 to 78 years, female = 122) completed the online task, including 6 children (<18 years). The experiment was conducted according to the guidelines of the Declaration of Helsinki, and approved by the Institutional Review Board of INSERM.

### 5.3. Procedure

To investigate whether the choice of strategy was amenable to instruction, after one block without instruction, subjects were instructed to either prepare their response as soon as the first stimulus appears (“anticipate” rule) or refrain from preparing any response before the second stimulus appears (“wait and see” rule). The rules were randomly assigned to participants before the second block of trials.

The task was developed using the Psychopy toolbox of Python and the Pavlovia website to store and share the experiment (https://pavlovia.org (accessed on 1 February 2023)). Participants had to first complete general information (pseudo, age, gender, presence of another person in the room during task completion). They then performed two blocks of the numerosity task (before and after they received the rule), each comprising 64 trials and lasting approximatively 6 min. A total of 118 participants were randomly assigned to the “anticipate” rule, whereas the remaining 118 participants were randomly assigned to the “wait-and-see” rule. Only the participants who completed the entire task were included in the results. In order to promote participants’ engagement during the task, we created a “pirate scenario” where participants were asked to recover as many treasure coins as possible (the dots displayed during the task were the treasure coins). Auditory feedback emitted after each trial with an incorrect or slow response warned the participant that “you lost these coins”. Positive feedback concerning progress in discovering the treasure was provided at the end of the first block to encourage participants to complete the second block.

### 5.4. Analyses

As in Experiment 1, descriptive statistics were conducted on the correct RT distributions at the group and individual levels. We classified, separately for each block of trials, participants as fast or slow responders relative to the group trough latency. For each of the two subgroups defined in the first block (“initially-slow” and “initially-fast”), we investigated response profile changes during the second block to determine whether the instruction produced abrupt RT gains or losses indicative of a strategic shift. We then conducted chi-square tests to investigate proportion changes according to the response profile for each part (fast–fast, fast–slow, slow–fast, slow–slow) and the rule (“anticipate”, “wait and see”). We conducted supplementary chi-square tests by separating the two blocks (before and after the rule) using the response profile and rule factors. Additional post hoc comparisons were conducted when needed. Cramer’s V was calculated to estimate the effect sizes for each chi-square test. 

We also analyzed the percentage of correct responses before and after instruction (Part1, Part2) using a 2 × 4 repeated-measures ANOVA with rule (“anticipate”, “wait and see”) and profile pattern (slow/slow, fast/fast, fast/slow, slow/fast) as between-subject factors. As the percentage of correct responses was not normally distributed, we log-transformed the data before conducting the ANOVA in order to fit them with our negatively skewed distribution: log-transformed %correct = log10 (max (%correct + 1) − %correct). Effect sizes were reported as partial eta-squared values (η_p_^2^).

Finally, RT distributions of incorrect responses were explored in Appendix C, using the same analyses as described above, in order to investigate if the process underlying the strategies was the same whatever the outcome of the response.

## 6. Results—Experiment 2

### 6.1. Descriptive Statistics

The group RT distributions (Figure 8A) were bimodal both before (purple distribution) and after (green distribution) the instruction nudge (rule). The two peaks stood 600 ms apart, with the first peak latency being around 500 ms and the second peak latency near 1100 ms. In other words, subjects behaved at home in the exact same way as in the lab during both neuroimaging (Experiment 1) and behavioral assessment (Tricoche et al.’s (2021) [21]. Individual RT distributions (Figure 8B) show that, before the rule, the proportions of fast and slow responders were roughly even, showing the same fast/slow 55/45 ratio as in Tricoche et al. (2021). After the rule, however, the first peak increased (at the expense of the second one; Figure 8A) as the proportion of fast responders increased up to a 71/29 dominance akin to that characterizing in-scanner Experiment 1.

Figure 9 illustrates the consequences of the instruction nudge according to its message, “anticipate” or “wait and see”, and subjects’ initial response profile, fast or slow. Figure 9A shows that both rules influenced subsequent group RT distribution, but the “anticipate” rule did so much more than the “wait and see” rule. Figure 9B shows that both response profiles displayed changes in group RT distributions after the instruction, but slow responders were more affected than fast responders. Note that this pattern characterizing RT for correct responses was also found for incorrect responses (Appendix C).

In order to validate these qualitative observations, we statistically investigated through chi-square tests the proportion of fast and slow responders according to the rule or the part of the experiment.

### 6.2. Analysis of RT Distributions

Participants were divided into four subgroups (Table 1): two changing their response profile between the two parts of the experiment (fast–slow, slow–fast) and two keeping their response profile (fast–fast and slow–slow). This classification showed that the majority of subjects (74.2%) simply kept the same profile throughout the two parts of the experiment, whether it was in agreement (38.2%) or in conflict (36.1%) with the intervening instruction. Yet, one out of four subjects (25.8%) did shift from their initial profile to the other for the second part of the experiment, and there were twice as many subjects shifting when instructed (17.2%) than subjects shifting spontaneously (8.6%).

Specifically, 49/107 (44%) initially slow individuals responded more quickly in block 2, with the majority of them doing as instructed (30/49 i.e., 61% of them); 11/126 (9%) initially fast individuals responded slower in block 2, virtually all of whom acted in compliance with the instruction (10/11 i.e., 91% of them). These findings indicate that instruction-driven response shifts did occur, leading to abrupt rather than continuous RT changes, generally gains, but losses too. Participants in the fast–slow subgroup were rare, but their presence demonstrated that a brisk on–off 600 ms RT loss was possible following a mere instruction.

The chi-square test indicated a significant profile pattern x rule interaction (X^2^ = 10.14, *p* = 0.02, V = 0.12), with the effect size categorized as small, and post hoc analyses disclosed significant results only for the (less represented) fast–slow subgroup, revealing that the proportion of fast responders during Part 1 becoming slow responders during Part 2 was higher for the “wait and see” rule (4.2%) than the “anticipate” rule (0.4%). Therefore, fast participants became slow responders only if specifically instructed to do so. Interestingly, a chi-square test with rule and response profiles as factors conducted in participants who changed their response profile between the two parts (fast–slow and slow-fast) was significant (X^2^ = 7.8, *p* = 0.005, V = 0.10), with an effect size categorized as small, showing that the number of participants for each response profile was different according to the received rule. Thus, initially fast responders who changed their profile to slow were more likely to have received the “wait and see” rule than the “anticipate” rule. Conversely, initially slow responders who changed their profile to fast were more likely to have received the “anticipate” rule than the “wait and see” rule.

We then conducted two chi-square tests on Part 1 and Part 2 separately, as represented in Table 2. We did not find any significant effect of rule on Part 1. This was expected because, during this part, participants had not yet received any rule (X^2^ = 0.61, *p* = 0.43, V = 0.03). During Part 2, the effect of rule on the proportion of fast and slow responders did not reach significance (X^2^ = 2.95, *p* = 0.09, V = 0.06). Finally, we asked participants to rate their success in following the rule during the second part between 0 (not at all) and 10 (totally). We found a similar mean score across subgroups, whatever the received rule (“anticipate”: fast–fast = 6.13, fast–slow = 6, slow–fast = 6.92, slow–slow = 5.26; “wait and see”: fast–fast = 6.52, fast–slow = 6.75, slow–fast = 6.14, slow–slow = 6.23). This suggests that the subjective rating scores were low in all subgroups and did not predict the profile participants adopted following the rule.

### 6.3. Analysis of Percentage of Correct Responses

We conducted a repeated-measures ANOVA on the percentage of correct responses before and after the rule was given (Part 1 and Part 2) using the rule (“anticipate”, “wait and see”) and profile pattern (fast/fast, slow/slow, fast/slow, slow/fast) as between-subject factors. We found a main effect of rule (F (1228) = 16.13, *p* < 0.001, η_p_^2^ = 0.07), indicating that participants became less accurate after receiving the “anticipate” rule (Part 1: 86.47%, Part 2: 79.61%) compared to the “wait and see” rule (Part 1: 87.18%, Part 2: 87.25%). We also found a main effect of profile pattern (F (3228) = 2.70, *p* = 0.05, η_p_^2^ = 0.03). Pairwise comparisons indicated a significant difference between the slow responders who stayed slow during Part 2 and participants who were fast during Part 2 (fast–fast: *p* = 0.009 and slow–fast: *p* = 0.02) (Figure 10). No difference was found between fast–slow and slow–slow subgroups. However, we did not find a significant rule × profile pattern interaction. These results suggest that participants were globally less accurate after receiving the “anticipate” rule and that participants who were or became fast during Part 2 were less accurate than consistently slow responders.

## 7. Discussion—Experiment 2

In this second experiment, 236 participants, mostly adults, completed the numerosity task online. During the first block of trials, they performed the task without any specific rule, whereas just before the second block, they received, randomly, one of the two following instructions: (1) try to prepare your answer as soon as the first stimulus appears (“anticipate” rule); (2) try not to prepare your answer before the second stimulus appears (“wait and see” rule). Subjects showed in the two parts of the experiment bimodal RT distributions with the two peaks standing 600 ms apart. In other words, subjects responded at home following the exact same dual mode as that seen in the lab during both neuroimaging (Experiment 1) and behavioral assessment (Tricoche et al. 2021) [21]. The main finding, however, was that about one-quarter of the participants (26%) showed a switch in their response profile in the second block of trials, the majority of whom acted in compliance with the received instruction (even if the cost was some accuracy loss). This demonstrates that a mere instruction can lead to an on–off 600 ms RT gain or loss, a fact more likely to reflect the brutal switch between two irreconcilable, mutually exclusive strategies, rather than the smooth continuity of gradual processes such as strategy honing, incremental learning or response automatization [49].

## 8. General Discussion

Together, Experiments 1 and 2 showed over four different sequential comparison tasks that two strategies can co-exist in participants. These strategies had different speeds and sometimes also differed in accuracy. That is, responding fast could impair accuracy, particularly in children and after being instructed to anticipate response selection. These results fit with predictions of the traditional decision-making models with a speed–accuracy trade-off [11,35], which seems here to disappear during adulthood. Whether the present findings can be generalized to tasks where S1 and S2 appear simultaneously rather than sequentially remains to be determined, which could further highlights the link between anticipation and the fast strategy.

### 8.1. Strategy Preference Is Influenced by Age and Environment

We found that, within each task, the vast majority of participants stuck to the same strategy, as disclosed by their unimodal individual RT distributions. Subjects’ preferences were also generally stable across all four tasks, with such consistency present in 90% of the adults and about 70% of the children and adolescents. In addition to its remarkable stability within subjects, strategic control was also strikingly ubiquitous across situations as dissimilar as the in-scanner testing of Experiment 1 in this study, the behavior-only lab testing of Tricoche and collaborators (2021), and the at-home online testing of Experiment 2. Barring a few rare exceptions in the youngest group, strategic segregation across subjects was a behavioral invariant in this study and previous ones.

Strategic control can nevertheless be modulated by factors including age and environment. Indeed, the proportion of fast responders increased with age along with, presumably, the refinement of anticipatory capacities over development. The fast strategy gains preeminence in early adolescence, albeit not quite reaching adult level. Although we demonstrated earlier that children can adopt the fast strategy as early as 8 years of age (they do it when observed by a peer; see below), our current results suggest that they may not do so as spontaneously or systematically as adults. In any case, the present study is, to our knowledge, the first to unravel the developmental trajectory of anticipation by tracking response strategies through childhood, adolescence and adulthood.

Regarding environmental factors, combining this study with the one of Tricoche and collaborators (2021) highlights two modulating factors: the social context and the experimental context. First, we found in a previous study that social context may influence the strategy used. Indeed, this study showed that, in both adults and children, the mere presence of a peer increased the likelihood that the participant behaves as a fast responder. Second, experimental context may play a role as well. Experiments 1 and 2 did not show the same proportion of fast and slow responders in adults. The 50-50 ratio seen in the first part of Experiment 2 fully replicates the balanced ratio seen in our previous (behavior only) study [21]. Such was not the case with the massive predominance (around 90%) of fast responders in the adults of Experiment 1. Data from Experiment 1 were collected during fMRI scanning, which, unlike behavior-only testing, means loud noise, forced immobility, cramped quarters and, often, several experimenters hovering nearby. Previous studies showed that this challenging environment influences the performance of participants, and particularly their RT [43,44,45]. Because fMRI scanning dissuades anxious or claustrophobic volunteers [46], it might lead to a sampling bias favoring high performers and fast responders. This could explain the higher proportion of fast responders in Experiment 1 compared to both Experiment 2 and our previous behavioral study [21].

Also influencing strategy selection were the instructions favoring either reactive or anticipatory responses. Experiment 2 was designed to specifically manipulate this context between the two phases of testing. It showed that instruction did nudge some subjects into a strategic shift entailing abrupt 500 ms RT changes.

### 8.2. Underlying Cognitive Processes and Neural Substrates

Using the diffusion model, we showed that differences between fast and slow responders occurred mainly at a non-decision time, being similar for all tasks and ages, suggesting a general process. Non-decision time aggregates the time dedicated to S1 processing and the time dedicated to preparing the chosen motor response. It was decreased to allow fast responses. These results, replicating our previous study [21], are in line with an anticipatory origin, being particularly adapted to sequential judgments where stimuli are presented successively. Moreover, by manipulating the instruction, we succeeded in nudging some of the online participants into abrupt massive RT changes that rule out non-strategic explanations of the segregation between slow/fast responders, such as gradual learning or automatization processes. We did find a modulation of the decision threshold parameter, but only for adults and only in the multiplication and subtraction tasks, suggesting that a reactive process could contribute as well to optimal strategic control in terms of both speed and accuracy.

Based on our results, it seems that children did not spontaneously adopt the most adaptive anticipatory strategy, even when the tasks allowed it. The fast strategy, which enables participants to deal with risk early, might increase working memory load relative to the slow strategy. Therefore, it is possible that participants opting for the fast strategy might have higher working memory skills than participants adopting the slow strategy. This hypothesis of a memory load difference between strategies remains to be investigated, but it is similar to that proposed by Chen, Hale and Myerson to discriminate between their slow and fast responders [48]. Because children have reduced working memory abilities [41], they might naturally avoid using the faster strategy and refrain from processing uncertainty, and rather wait until the second stimulus appears to make their choice.

As the present data were collected in the context of several fMRI studies, it is also interesting to consider that the different strategies uncovered here might be associated with different neural substrates, which might also depend on age. For instance, if our hypothesis is correct, we could find differences between fast and slow responders by measuring their working memory abilities and anticipatory capacity. At the brain level, contrasting the MRI functional data between fast and slow responders should reveal enhanced activity in brain regions related to anticipation of motor responses or to memory, such as prefrontal areas, as well as the thalamus, which was previously identified as a key node of anticipatory attention [50,51].

### 8.3. Methodological Consideration

The present study highlights the importance of analyzing the full RT distributions in psychophysic studies. Specifically, analyzing the individual RT distribution in all participants can reveal different subpopulations in the tested sample [34]. More importantly, it provides much more information than standard group-level measures such as the mean RT [52]. For example, in Experiment 1, the mean numerosity RT in 12 to 15-year-olds (968 ms) was clearly not representative of the whole group performance. In fact, it was close to the trough of the distribution (846 ms), i.e., to the lowest RT density, totally missing the high RT densities corresponding to both the fast strategy (around 575 ms) and slow strategy (around 1125 ms).

## 9. Conclusions

An examination of RT distribution in a large cohort of children, adolescents and adults performing four different two-alternative RT tasks with sequentially presented stimuli revealed an age-related increase in the proportion of fast responders with a mean RT of around 500 ms relative to slow responders with a mean RT of around 1100 ms. Asking (adult) subjects to anticipate their response produced in a proportion of them a similar 600 ms RT change when compared to subjects asked to refrain from doing so, supporting the idea of an abrupt strategy shift rather than a gradual process such as learning. Together, these findings suggest that sequential comparative judgments provide a reliable window into anticipation abilities, their improvement with development and adjustment to the environmental context. In addition, the present study emphasizes the need to take into consideration the full RT distributions in behavioral, but also neuroimaging, studies as mean group measures might conceal important individual differences.

## Figures and Tables

**Figure 1 behavsci-13-00646-f001:**
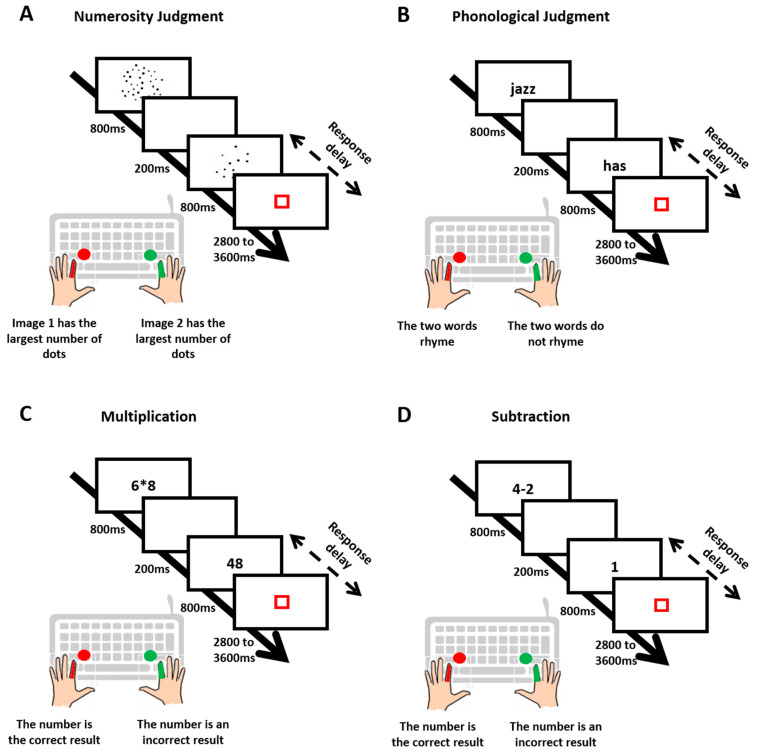
The trials sequence in the four tasks: numerosity judgment (**A**), phonological judgment (**B**), multiplication (**C**) and subtraction (**D**).

**Figure 2 behavsci-13-00646-f002:**
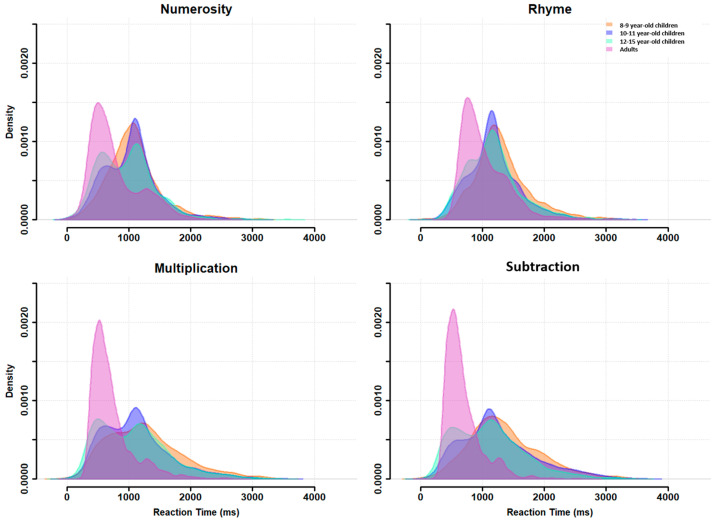
RT distributions at the group level for each task (panels) and each age subgroup (colored distributions).

**Figure 3 behavsci-13-00646-f003:**
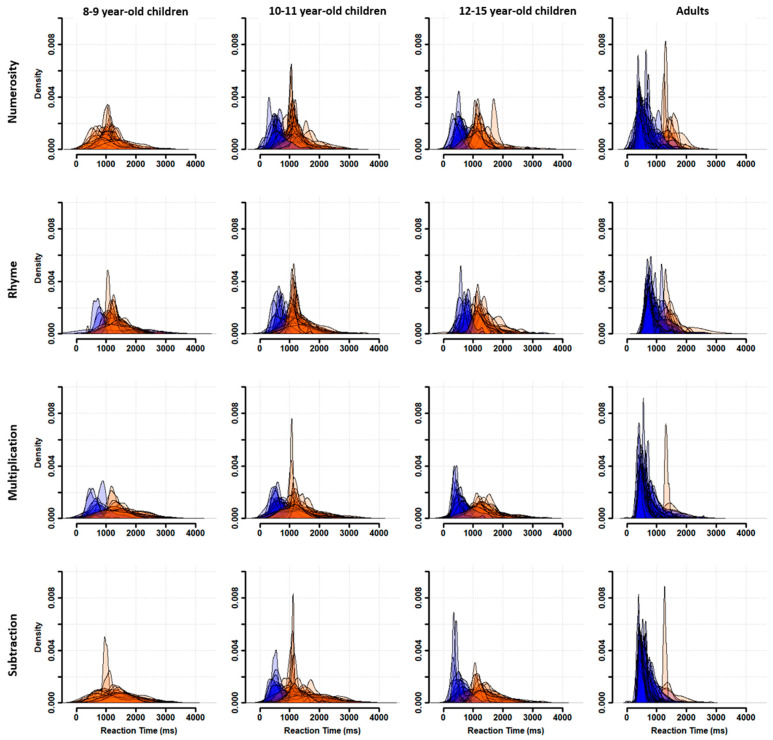
RT distributions at the individual level for each task (rows) and each age subgroup (columns). Participants were classified as fast (blue distributions) or slow (orange distributions) responders.

**Figure 4 behavsci-13-00646-f004:**
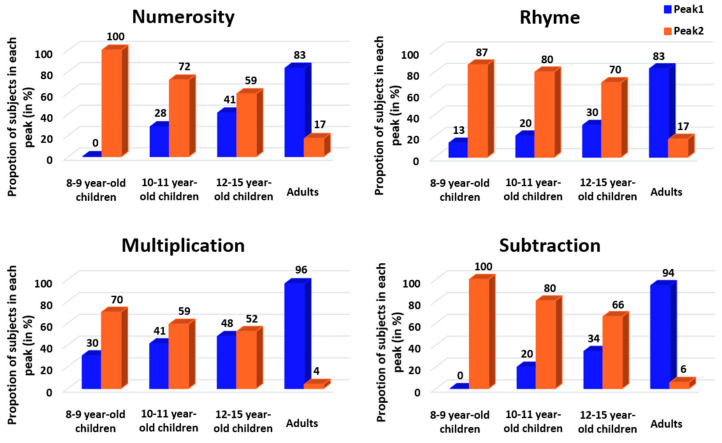
Proportion of participants in each peak (fast or slow responder) according to the tasks (panels) and age subgroups. Blue = fast responder (peak 1), orange = slow responder (peak 2).

**Figure 5 behavsci-13-00646-f005:**
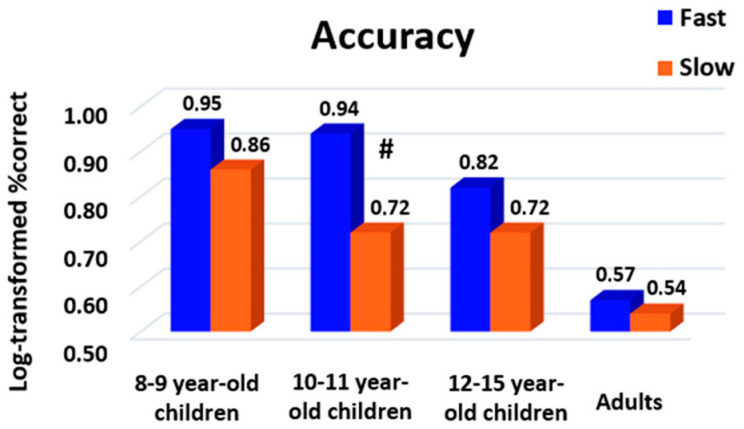
Accuracy, expressed in log-transformed percentages of correct responses, of participants for each peak (fast or slow responder) according to the age subgroup. A high value indicates a lower level of performance. Slow responders were more accurate than fast responders, particularly in 10–11-year-old children, suggesting a speed–accuracy trade-off which decreases with age. Blue = fast responder (peak 1), orange = slow responder (peak 2). **#**: *p* = 0.09 with Bonferroni correction and *p* = 0.02 with FDR correction.

**Figure 6 behavsci-13-00646-f006:**
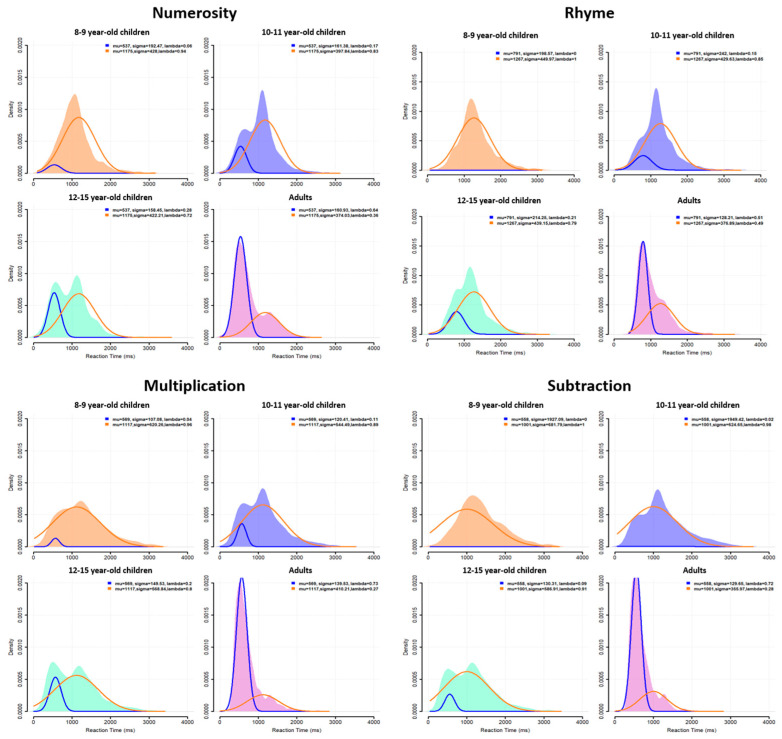
Mixture model fitted to the group-level RT distributions for each task and age. Blue curve corresponds to fast responders, orange curve corresponds to slow responders, as estimated by the model. Distributions on the back correspond to the actual group distributions as plotted in Figure 2.

**Figure 7 behavsci-13-00646-f007:**
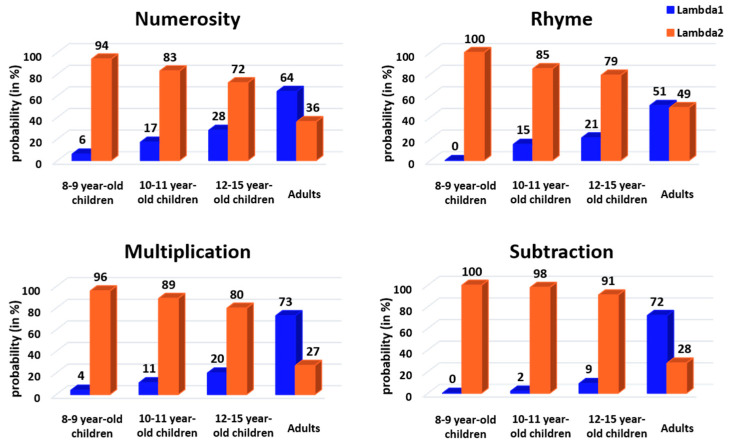
Mixture model lambda values (i.e., probability in each age group and task that a given RT taken randomly from the actual distribution belongs to the fast or the slow responder fitted Gaussian distribution) plotted separately for age groups and tasks. Blue = fast responder (Lambda 1), orange = slow responder (Lambda 2).

**Figure 8 behavsci-13-00646-f008:**
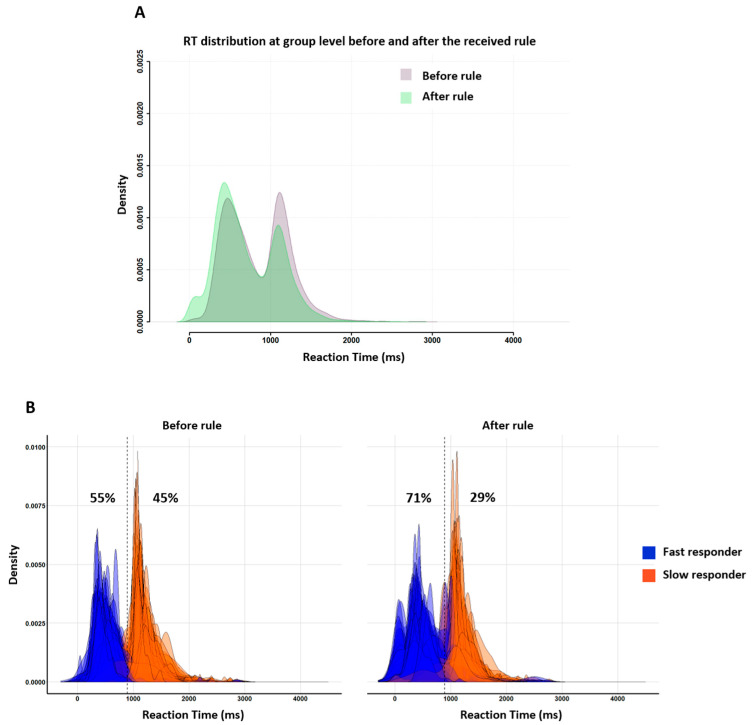
Superimposed RT distributions between the first (before rule, purple distribution) and the second blocks of trials (after rule, green distribution) for all participants, whatever the received rule (**A**). Individual RT distributions before (**left**) and after (**right**) the rule. (**B**) Participants were classified as fast (blue distributions) or slow (orange distributions) responders. Note the 16% increase (55 to 71%) in the proportion on fast responders in the second part of Experiment 2.

**Figure 9 behavsci-13-00646-f009:**
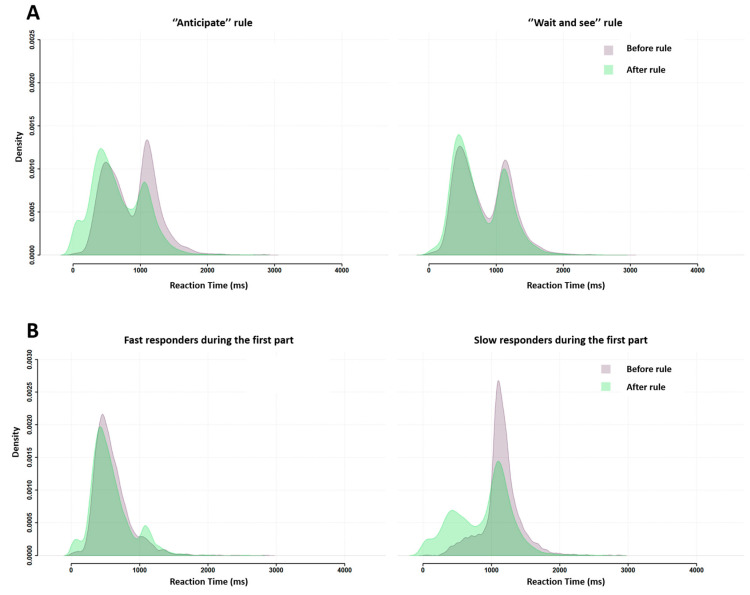
Superimposed RT distributions before (purple distribution) and after (green distribution) subjects were asked to either “anticipate” (**left**) or “wait and see” (**right**) for all subjects together (**A**) or separately for each response profile during the first part of the experiment (fast responders on the left, slow responders on the right) (**B**).

**Figure 10 behavsci-13-00646-f010:**
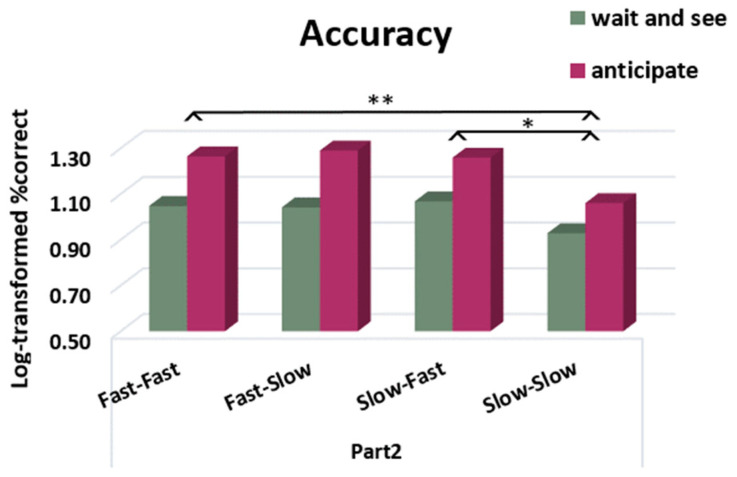
Accuracy after receiving the rule (Part 2), expressed in log-transformed percentages of correct responses, according to the “anticipate” or “wait and see” rule and the profile pattern between Part 1 and Part 2 (fast–fast, fast–slow, slow–fast, slow–slow). Participants were less accurate after receiving the “anticipate ” rule, and participants who were or became fast during Part 2 were less accurate than consistently slow–slow responders (*: *p* < 0.05, **: *p* < 0.01).

**Table 1 behavsci-13-00646-t001:** Number (N) and proportion (%) of participants according to their response profile (slow or fast responder) in Part 1 and Part 2 and the rule they received (“wait and see” or “anticipate”). “Fast-Fast” and “Slow-Slow” are participants who did not change their response profile during the entire experiment, whereas “Fast-Slow” and “Slow-Fast” are participants who did change their response profile.

		Profile in Part 1—Profile in Part 2				
		Fast—Slow	Slow—Fast	Fast—Fast	Slow—Slow	Total
Rule	“Wait and see”	N = 10 4.2%	19 8.1%	58 24.6%	31 13.1%	118
	“Anticipate”	1 0.4%	30 12.7%	60 25.4%	27 11.5%	118
	Total	11	49	118	58	236

**Table 2 behavsci-13-00646-t002:** Number (N) of participants for Part 1 (top) and Part 2 (bottom) separately according to their response profile (slow or fast responder) and the received rule (“wait and see” or “anticipate”).

Part 1 Only (before Rule)		Response Profile		
		Fast Responders	Slow Responders	Total
Rule	“Wait and see”	N = 68	50	118
	“Anticipate”	61	57	118
	Total	129	107	236
**Part 2 only** **(after rule)**		**Response profile**		
		**Fast responders**	**Slow responders**	**Total**
Rule	“Wait and see”	N = 77	41	118
	“Anticipate”	90	28	118
	Total	167	69	236

## Data Availability

Datasets for Experiment 1 are available on OpenNeuro (Berteletti et al., 2021; Booth et al., 2020) [27,28]. Datasets for Experiment 2 are available on the Open Science Framework website at https://osf.io/b54xv/ (accessed on 1 February 2023).

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
