# Peer review of "Developmental Trajectory of Anticipation: Insights from Sequential Comparative Judgments"

_behavsci, 2023, doi:10.3390/bs13080646_

Round 1

Reviewer 1 Report

The authors describe their research investigating the distribution of response times in decision making, where they mainly focus on the possibility of a bimodal distribution. They describe two experiments, a first in which they investigate differences between age groups and a second one in which they investigate changes in strategy as a result of strategy instruction. In this reviewers opinion, the article shows promise but a number of points will need to be addressed to warrant publication.

Introduction

-          Please add a description in the introduction of what the different strategies are supposed to represent (what cognitive of emotional/motivational processes) and what the relevance is of the ability to distinguish between them.

-          The description of the research questions and hypotheses is unclear to this reviewer, please describe the research questions and hypothesis in more detail so that they can be related to the chosen methods and analyses.

Methods exp 1

-          Overall the design and administration procedure should be clearly stated, so that there is also a picture of possible confounding factors (such as tests being made in a scanner vs online at home). Did fatigue or guessing possibly play a role, and did you take that into account somehow? Was the test administration somehow counterbalanced to account for sequence effects? How long did the test administration take?

-          With the description of the used instruments, please describe what they aim measure, and provide information on their reliability and validity.

-          If available, it would be useful to provide background information about participants’ cognitive abilities, SES, etc.

-          If you use a mixture model, please explain in your method section what exactly a mixture model is, what questions it can answer and what it’s parameters mean.

-          Please describe more thoroughly how exactly participants were scored as slow or fast responders and why that scoring was chosen.

Results exp 1

-          In general the analyses now seem disconnected from the research questions, making it unclear what question the authors are trying to answer with the analyses they describe. Please provide a more thorough description of the questions each analysis is trying to answer/hypothesis it is testing.

-          Please provide descriptive statistics, also for accuracy, so that it is clear what the average accuracy levels are for the different tasks, ages, and difficulty levels.

-          For 3.3 it is unclear to me what the within-subjects and what the between subjects factors are. In addition, how a general profile score was made over all tasks was not described, which makes it difficult to interpret what that score actually means. The addition of the tasks as, what I assume is a within-subjects factor, also makes effects less clear. I would think a separate analysis per task, with the profile determined per task, would be more informative, especially in the light of children’s inconsistency in response times over tasks.

-          For 3.3, please provide effect sizes.

-          For 3.4, the added value of the mixture model is not made clear, and unclear to this reviewer what it adds to answering the research question.

-          Difficulty was mentioned in the methods, but if I understand correctly, not included in the analyses. Task difficulty/complexity could be a differentiating factor, as previous research showed (e.g. Yulia A. Dodonova, Yury S. Dodonov, Faster on easy items, more accurate on difficult ones: Cognitive ability and performance on a task of varying difficulty, Intelligence, Volume 41, Issue 1, 2013, Pages 1-10, https://doi.org/10.1016/j.intell.2012.10.003. Goldhammer, F., Naumann, J., Stelter, A., Tóth, K., Rölke, H., & Klieme, E. (2014). The time on task effect in reading and problem solving is moderated by task difficulty and skill: Insights from a computer-based large-scale assessment. Journal of Educational Psychology, 106(3), 608–626. https://doi.org/10.1037/a0034716; Naumann, J., & Goldhammer, F. (2017). Time-on-task effects in digital reading are non-linear and moderated by persons’ skills and tasks’ demands. Learning and Individual Differences, 53, 1–16. https://doi.org/10.1016/j.lindif.2016.10.002). Could this be included in the analyses?

Discussion exp 1

-          The authors discuss that a certain strategy has not developed in younger children (4.2), isn’t fast or slow response just level of automatization? How does automization of the basic cognitive abilities that are necessary to make the decision (provide a correct answer) play a role here? If automization is a factor, what does that mean for the meaning of these strategies?

-          P12 r 364, the instability/variability would be in line with the expectations based on Siegler’s (1996) overlapping waves theory, where stability might even signal learning. Please also include this view in the discussion, as the changes in efficiency of strategy use may play a role here.

Methods exp 2 (analyses)

-          Please also indicate the distinction between “within” and “between” factors in the ANOVA analysis. Is it a Repeated Measures ANOVA?

Results exp 2

-          Please provide effect sizes for the effects, with an interpretation, and also provide information on the direction of each effect in the text.

Discussion

-          7.2 It seems to this author that decision making is not the only ability that plays a role here, but (especially for children) there is a factor of (simple) problem solving, and the cognitive abilities involved in that influence the strategies used (and the strategies used may therefore be influenced by experience with similar task and learning that has differentially taken place).

-          7.2 Since only correct response times were taking into account, but there seems to be a 50% chance of guessing correctly, what role could a guessing strategy have played?

-           7.4 what (cognitive) mechanism could be behind the differences in strategy and bimodal distribution? Working memory is offered, but does not seem to be an explanation for a bimodal distribution, as working memory capacity is not dichotomous, so would more likely result in a more even distribution

English language writing is fine

Reviewer 2 Report

Abstract
Lines 26-27: It is not clear why it its needed assess decision-making strategies for a naive reader.

Introduction
Line 42: Why not use the Drift Diffusion Model (DDM)? There are numerous articles investigating individual differences pertaining to the parameters in the DDM:
https://doi.org/10.1016/j.cognition.2011.02.002
https://doi.org/10.1016/j.tics.2016.01.007
https://doi.org/10.1037/xge0000774

Note that the last reference (Lerche et al.) made thorough investigation of how different parameters in the DDM related certain parameters to different IQ-measures which may be the underlying differences you found in the simple RT-distributions.

Lines 61-68: Again, I ask myself, haven't this already been mapped out in Lerche et al.'s paper, but using a different modelling strategy. At least you need to acknowledge that there similarities and possibilities that you are measuring the same thing (e.g., IQ) with RT.

Material and methods
Lines 94-96: Were participants completing the random or counterbalanced order?

Analyses
Lines: 162-164 What is the mixture model? I might not be the target audience, but it would be nice with a short description of what the mixture model is here. When googling mixture models I understood that lambda was a parameter in such models, and therefore I assume that the CMH lambda^2 test is a model-check (?) for such models. Provide more details or include an Appendix describing the model for the naive or interested reader.

Results
Lines 177-178: Here the reader learns that CMH lambda^2 tests were used to classify evolution of RT over age, this information could/should have been given in the analyses since it seems to be central in your analyses.

Lines 243-244: Was percentage correct responses normally distributed?

Lines 263 - 272: This paragraph would have been much easier to digest if the mixture model was introduced earlier on.
Line 274: What does lambda signify?

Discussion E1
Lines 360-361: But were they answering as accurate as when their responses were slower? This is crucial, because you can always answer fast, but not fast and accurate.

Discussion E2
Lines 615-619: What is the function of knowing what distribution the participants belong to? You give some hints to the question in 7.4, but I think this needs to be clearly stated at the beginning of the paper.

General comments
Personally, I prefer short discussions after each experiment and a more detailed general discussion of all experiments at the end. The risk is that the reader does not remember what has been said in the discussion of the first experiment when she reaches the final discussion.

As I said earlier, the studies need to be justified in relation to previous DDM-literature and the overarching goal of different RT-distributions.

There are a few spelling errors an grammatical mistakes - please proof read once again.

Round 2

Reviewer 1 Report

The authors have satisfactorily addressed all comments in depth. 

Quality of English seems sufficient

Author Response

We thank the Reviewer 1 for this second round.

We have re-read the manuscript in depth and made minor corrections, we confirm that each comment and concern of Reviewer 2 has been addressed, and the number of auto-citations is now under 10% (5/52 =9.6%) as requested. 
